# Contesting Religious Boundaries with Care: Engaged Buddhism and Eco-Activism in the UK

**Zoe Zielke**

School of Anthropology and Museum Ethnography, University of Oxford, Oxford OX2 6PE, UK;
zoe.zielke@keble.ox.ac.uk

**Abstract:** The word "Buddhism" conjures up a variety of images and connotations: monks meditating on hilltops, mindfulness, cheerful Buddha caricatures. It is unlikely that these depictions suggest engagement with societal issues. And yet, this is precisely what many Buddhist communities and traditions are involving themselves in around the world. Often referred to as "engaged Buddhism", this development in the Buddhist tradition refers to the application of Buddhist principles and practices to situations of social and environmental suffering. Nevertheless, there are critics of this emerging trend who contend that Buddhists should refrain from engaging in societal issues, believing that such involvement contradicts the teachings of the Buddha and distracts from the ultimate goal of liberation. Built on two years of ethnographic research, this paper explores the ways in which a particular environmentally engaged Buddhist group known as "Extinction Rebellion Buddhists" adapt their religious beliefs and practices in response to the challenges posed by the Anthropocene, where concerns for our collective world have resulted in increasing interest in the ways in which humans actively care for the environment. In reformulating Buddhist principles and meditation as a "politics of care", care becomes a tool for change, with the group not only confronting the pressing issues of the Anthropocene but also disrupting Buddhism's traditionally inward-looking, other-worldly tendencies, carving out space for autonomy and transformation within the broader landscape of UK Buddhism.

**Keywords:** engaged Buddhism; Buddhist modernism; United Kingdom; care; Buddhist ecology; activism; embodiment

## 1. Introduction

"No more coal! No more oil! Keep our carbon in the soil!"; the chanting of these words emanated around Parliament Square, where thousands of protesters had gathered on 1 September 2020. Soon, their voices became drowned out by the intense drumming of a samba band. Colorful smoke bombs shot upwards towards the unusually blue London skyline above, and neon banners billowed in the breeze, most of them emblazoned with an encircled hourglass smeared in black paint. This is the trademark symbol of "Extinction Rebellion" (XR) (Figure 1), a UK-based environmental movement which uses nonviolent "mass civil disruption" (blocking roads, risking mass arrest, staging protests, or, as they call them, "Rebellions"), to emphasize the urgency of the ecological crisis and protest the lack of government action in protecting the environment from the growing impacts of climate change. Identifying as a "decentralized" organization, as a whole, XR serves mainly as a way to share information, with smaller subgroups created based on factors such as geographic location, ethnicity, occupation, and religion. These groups are involved in instigating actions themselves, becoming involved in actions with other subgroups, participating in the movement-wide Rebellions, which often occur twice a year, and developing direct action tactics which adhere to their groups' mentalities. On this day, the organization has come together as a collective to take over the surrounding area of Parliament, demanding recognition, attention, and change from those high up in the UK government.

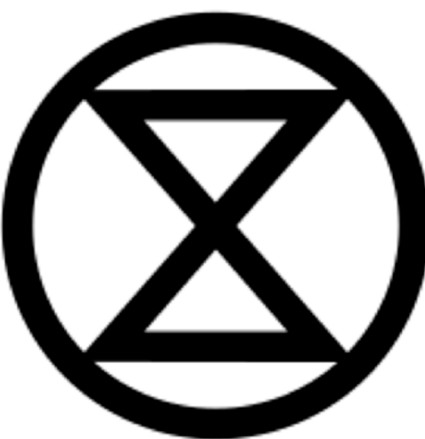

**Figure 1.** An encircled hourglass. The symbol for Extinction Rebellion.

However, I am not here for the samba band, nor the speeches being read out over a loudspeaker, not even to observe the group as a whole. My focus is drawn to a small group of fifteen individuals, sitting in a line across a road that runs parallel to Parliament Square. They are blocking off traffic, much to the dismay of commuters and taxi drivers, but first and foremost they are meditating. Quiet, poised, and calm, the group remains in a meditative state for hours on end while chaos ensues all around them. Through the pandemonium emanating from the noise of the demonstrators, the tensions between the police and the protest, and the London traffic, with many individuals expressing anger at their daily commute being disrupted, the group appears completely unphased. Suddenly, a blockade of uniformed police officers forms a parallel line alongside the group (Figure 2). I watch on as officers stoop down to the level of the protestors, informing them that if they do not move, they will be detained. Their words have little effect. The protestors do not respond, nor do they even acknowledge the existence of law enforcement. Their eyes remain closed, their positions remain fixed, and their breathing remains slow and deep. After a further three warnings given by police I watch on as, one by one, members of the group are physically lifted and dragged into a nearby police van, still in their meditative state. All that remains of them by the end is a large turquoise banner which reads "XR Buddhists".

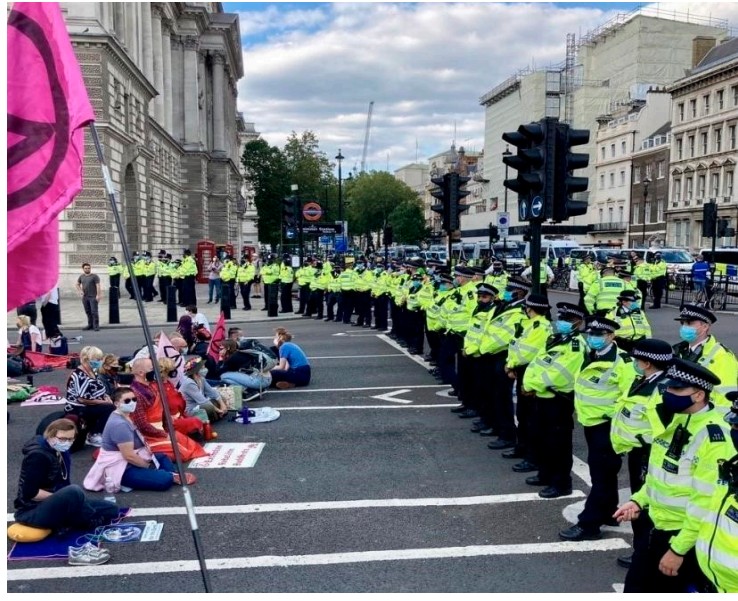

**Figure 2.** Photo by "Allison", XR Buddhists, 1 September 2020.

XR Buddhists are one of the many subgroups of Extinction Rebellion, implementing silent, seated meditation as a way to adhere to XR's emphasis on "mass civil disruption" and grapple with life in the United Kingdom in midst of the Anthropocene, an era marked by significant human impact on the Earth's ecosystems. In accordance with Paul Crutzen (2002), who is often credited with coining the term "Anthropocene", this research places this new planetary epoch as beginning in the latter part of the eighteenth century, when inventions such as the steam engine began to show the beginnings of growing global concentrations of carbon dioxide and methane. As this impact becomes ever more impressionable, the country has struggled with the complex challenges and consequences of climate change, biodiversity loss, and environmental degradation. The effects of industrialization, urbanization, and intensive agriculture are particularly pronounced in Britain's landscapes and ecosystems. Rising temperatures, changing precipitation patterns, and increased frequency of extreme weather events pose threats to both the natural environment and human communities. However, more than a scientific term, the Anthropocene is a political one, one that is meant to call attention to the changing and evolving notions of agency and responsibility regarding the state of the environment (Moore 2016).

Resulting from this increased notion of ecological accountability, intra-religious conflict has arisen amongst Buddhists in the UK regarding the role that their religion should play in politics and social change. While a large faction of Buddhists in the country continue to argue that their religion not only disagrees but explicitly advises against societal engagement, resulting in criticisms of Buddhism as being focused on "other-worldly pursuits" (Kuah-Pearce 2014, p. 27), as a result of colonialism, globalization, and widespread social and environmental catastrophes, developments have emerged that have prompted many practitioners to adapt to these growing complexities. Often referred to as "engaged Buddhism" ("applied Buddhism" or "Buddhist activism"), this development in the Buddhist tradition refers to the application of Buddhist principles and practices to situations of social and environmental suffering. This study contributes to the research on environmentally engaged Buddhism, specifically, and Buddhist modernism, more broadly, in the United Kingdom, with ethnographic data gained from interviews and participant experience conducted over two years in Bristol, Cambridge, and London between September 2020 and April 2023 amongst the group known as "Extinction Rebellion Buddhists" (XR Buddhists or XRB). From the onset of research, the study aimed to uncover how the group adapted their principles and practices to incorporate environmentalism and social activism when early interpretations of Buddhist scripture seemingly advise against such aspirations. It finds that (1) in drawing on Buddhism's lengthy history with care and well-being, for oneself and for others, XR Buddhists reformulate religious principles outside doctrinal standards as a politics of care, one which acts as an ethical imperative that encourages Buddhists to engage in the interconnected suffering of all life; (2) with meditation producing an embodied understanding of Buddhist tenets (see Cook 2010; Pagis 2010), the bodily sensations achieved in XRBs' actions work to formulate an intimate practice of care for the Earth which simultaneously engages with and raises awareness for the climate crisis; and (3) it is these associations between mind, body, and the external world achieved in the group's meditation-as-care which work to contest and challenge Buddhism's traditionally inward-looking inclinations, carving out space in the British landscape of Buddhism for autonomy and change.

## 2. Overview of Research Methods and Data

My involvement with XR Buddhists sprung largely from snowball sampling which began with the group's Facebook page where, in the summer of 2020, I first messaged the admin of the group, who I refer to as Robin. As a Pure Land Buddhist who lived in a Buddhist eco-commune near Malvern, religion and environmentalism dominated every aspect of Robin's life. She was a writer on Buddhist environmentalism as well as a psychotherapist and was heavily involved in nonviolent direct action. A cursory search of her name on YouTube revealed a plethora of actions she participated in. One such video

is entitled "[Her name]'s Rebellion of One". In this video, Robin sits alone, meditating on a busy street in London. She is dressed in all black and wears a sign on her back which reads "I'm terrified starving people will resort to violence because of the climate crisis". Onlookers stop and stare, some with their phones out filming and taking pictures, others laughing. A female police officer approaches her, grabbing her by the shoulder and attempting to pull her up into a standing position, but Robin remains still and seated. The police officer then resorts to a more aggressive tactic. She grabs Robin by her arm and starts to drag her out of the street, but Robin simply lays there, not reacting and continuing to meditate. The laughter from onlookers increases. After being moved onto the sidewalk and away from the oncoming traffic, Robin calmly and quietly readjusts her sign and continues her meditation.

Robin acted as somewhat of a guide for me during my initial stages of research, putting me in touch with other XR Buddhists, inviting me along to the September 2020 Rebellion, and being my first interview. Coming from an anthropological standpoint, I aimed to implement participant observation as my main research method, or "open-ended inductive long-term living with and among the people to be studied, the sole purpose of which is to achieve an understanding of local knowledge, values, and practices" (Howell 2018, p. 2). This was to be supplemented by structured and semi-structured interviews. However, very early on in my fieldwork, I decided to opt for a similar, yet alternative approach. This was inspired by a discussion I had with Robin at the start of research about potentially conducting participant observation at an upcoming action. She responded with "as long as you're not making anyone uncomfortable. . .in these actions a lot of personal things can come up, it would be great if you were sensitive to that. And it would be even better if you joined in yourself!" From this interaction, in addition to other experiences of XRBs' actions where I was consistently invited to participate, I quickly realized that the only way I would be fully accepted by the group was if I too became an XR Buddhist of sorts. Learning to meditate was central to this, leading me to the concept of "participant-experience" (Hsu 2006). Just as Hsu learned to be an acupuncturist during research, something which took effort, work, and conscious determination, I too learned to be a meditator and, from this, an XR Buddhist. In virtual meetings, and in-person guided meditations before actions, I joined along in sitting, closing my eyes, and meditating, fully engaging with the group and their exercises. From this came a kind of experiential knowledge where I was able to more deeply understand the experiences of my interlocutors: the struggles of the wandering mind, the difficulties making time for daily at-home meditation practice, the bonding and solidifying affects that meditation had amongst the group. In actions, I participated in all realms of activism. I meditated on busy roads, along streets, in patches of grass, in shopping malls. I also participated in media, snapping photographs, taking videos, and uploading them to the various XRB and Extinction Rebellion online forums. Sometimes, I even engaged in outreach, passing out flyers, speaking with passersby, attempting to initiate a dialogue about the climate crisis and why XR Buddhists are doing what they are doing. With such a close connection to the subject and group being studied, I did run the risk of developing biases and simply reproducing participants' points of view. However, by engaging in "participant-experience", my experiences of meditation and activism became interlocked with those of my interlocutors, enabling meaningful conversations about the diverse embodied sensations associated with meditation. Therefore, although I am sympathetic to the causes of XR Buddhists, I have attempted to approach this research with a critical eye and do not desire to romanticize or deify the movement.

*Demographic and Ethical Considerations*

Today, there are almost 200 Extinction Rebellion Buddhists nation-wide. Thirty audio-recorded interviews were conducted amongst the group, with members' names pseudonymized so as to guarantee anonymity with the increasingly harsh criminal sentencing occurring for direct-action activists. Interviewees reflected quite an even proportion of men versus women; however, disparities in the research did exist and were quite strong.



Overall, the vast majority of informants were at least middle-class, white, highly educated, and of middle to late age. Moreover, most participants found Buddhism later in life, meaning that their membership in Extinction Rebellion Buddhists sprung from their own individual convictions as opposed to familial or relational ties. Although the broader movement of Extinction Rebellion is quite diverse, Buddhist sanghas, engaged Buddhist groups, and, more broadly, Western Buddhism itself, is predominantly white (Vesely-Flad 2017). The decisive lack of presence of black Buddhists and Buddhists of color is something which has been researched by multiple academics, especially in regard to how the small minority of black and POC Buddhists engage in their predominantly white surroundings (see McNicholl 2018; Yetunde and Giles 2020). Unfortunately, there was not much that could be done to accommodate this in this research besides acknowledging that any outputs will reflect a decisive lack of voice from engaged Buddhists of color.

Moreover, another divide does exist in XR Buddhists, which is generational. With engaged Buddhism becoming popularized in the UK and the U.S. as a result of Buddhist responses to the Vietnam War, and continuing today, with the vast majority of my informants crediting the Vietnam War as initiating their interest in engaged Buddhism, this has resulted in the majority of XR Buddhists being well into their middle age. This brings up interesting questions regarding what multiple of my interlocuters have dubbed the "aging sangha". Buddhism, and religion in the West in general, is experiencing a sharp decline in membership, as younger people are becoming less and less interested. This phenomenon has become so severe that UK journalists have begun to write about what they have termed "Generation No-Religion". With the vast majority of my informants being above the age of 40, this information is especially interesting when looking at the age of XR Buddhists in relation to the wider movement of Extinction Rebellion, which possesses a large youth demographic. These findings pose questions of how XR Buddhists feel as an "aging sangha", what, if anything, they are doing to attract larger youth numbers, and how this reflects broader issues in Buddhism and religion as a whole. Unfortunately, my research did not allow me to interrogate such questions thoroughly, but it is apparent from multiple interviews where this phenomenon was raised, in addition to many Buddhist groups, whether they identify as "engaged" or not, beginning to create distinct "youth groups", that the severe lack of youth presence is seen as a substantial detriment, especially in XR Buddhists, where many informants have discussed with me the inspiration they have felt from the youth leadership in the broader environmental movement.

### 3. From Buddhist Modernism to XR Buddhists: Engaged Buddhism in the UK

The origins of engaged Buddhism are often traced back to the mid-20th century, when the term was first coined by Vietnamese Zen master Thich Nhat Hanh, who developed ideals which combined Buddhist principles with social engagement to protest the ongoing suffering caused by the Vietnam War. However, its practice can also be traced to other Asian countries such as Sri Lanka (see Queen and King 1996, p. 21), Burma (Sarkisyanz 1965), and Japan (Henry 2013, p. 101). In Tibet, for example, engaged Buddhism has a long and complex history, deeply intertwined with the socio-political developments and spiritual traditions of the region (see Cabezon 1996). The Ganden Phodrang period (17th to 20th century) saw the rise of the Dalai Lamas as both spiritual leaders and political heads of Tibet. During this time, the Dalai Lamas and other high-ranking lamas actively engaged in political matters, seeking to promote social welfare and protect the interests of the Tibetan people. However, it was the Chinese invasion and occupation of Tibet in the mid-20th century that drastically changed the landscape of engaged Buddhism in Tibet. The Dalai Lama and many Tibetans went into exile, leading to a significant disruption in the traditional monastic institutions and the spiritual leadership they provided. Engaged Buddhism took on new dimensions as Tibetans in exile sought ways to preserve their culture and advocate for their homeland's autonomy. Similarly, after venturing out on multiple missions to the U.S. and Europe to call for an end to hostilities in Vietnam, and with countless of his books banned to the public, Thich Nhat Hanh's outspokenness resulted in

his exile from his home country for thirty-nine years. His books went on to be published in America, which resulted in engaged Buddhism becoming popular in the west in the 1960s after civil society groups became inspired by Hanh and Buddhist ideals in terms of addressing violence, poverty, environmental destruction, etc. (Queen 2000). Hanh defines engaged Buddhism as[1]

> The kind of Buddhism that is present in every moment of our daily life...the kind of wisdom that responds to anything that happens in the here and the now—global warming, climate change, the destruction of the ecosystem, the lack of communication, war, conflict, suicide, divorce. As a mindfulness practitioner, we have to be aware of what is going on in our body, our feelings, our emotions, and our environment. That is engaged Buddhism.

Principles and activities connected to engaged Buddhism have been inculcated into groups, sanghas, and communities all over the world as a result of Thich Nhat Hanh's and other notable engaged Buddhists' work, such as the Dalai Lama and Dr. B.R. Ambedkar, among others. Engaged Buddhism, however, is a decentralized movement, one that is not tied to a specific location or reliant on the influence of a single charismatic leader. It can emerge wherever there are Buddhists who have the freedom and opportunity to actively participate in social and political matters (King 2009). Furthermore, engaged Buddhism is neither a new Buddhist sect nor part of an established sect. Therefore, although Theravada, Mahayana, Vajrayana, and nonsectarian Buddhists all may be involved in engaged Buddhism, not all Buddhists of any of these forms will identify as engaged Buddhists. According to Sallie King (2009, p. 2), engaged Buddhism is defined and unified "by the intention of Buddhists of whatever sect to apply the values and teachings of Buddhism to the problems of society in a nonviolent way, motivated by concern for the welfare of others and as an expression of their own Buddhist practices". With this characterization in mind, it is difficult to define who is and who is not an engaged Buddhist.[2]

Engaged Buddhism is thought to have emerged from the proliferation of what has been dubbed "Buddhist modernism". According to David L. McMahan (2008, p. 1), what many Americans and Europeans understand by "Buddhism" is, in fact, a hybrid of a number of Buddhist traditions that have interacted with the environment of the Western world, commonly referred to as "Buddhist modernism". To put it more concisely, Buddhist modernism, also known as "Protestant Buddhism" or "modern Buddhism", refers to the forms of Buddhism that began in the nineteenth century which combine Buddhist ideas and practices with key discourses of Western modernity (Bechert 1966).[3] Buddhist modernism is thought to have emerged from resistance to European colonization and the Christian missionization of peoples in Buddhist countries as well as an appropriation of Western philosophy, social forms, and ways of life (Morgan 2004). Both Asian Buddhist reformers as well as Western Orientalists shared the goal of presenting Buddhism as rational and harmonious with modern science to counter Western criticisms of the religion. This resulted in Buddhism being stripped of its "mythic and magic mysticism", with a focus, instead, on what reformers considered to be its fundamental teachings (Morgan 2004, p. 360). Buddhist modernists worked closely with Western scholars of Buddhism, cultivating new concepts that resulted from Western cultural influence. Therefore, it has long been agreed that Buddhist modernism was in conjunction and in response to colonialism, and this reforming spirit then acted as the catalyst for various movements such as engaged Buddhism.

As a result of Buddhist modernism, there has been a greater emphasis on texts, rationality, meditation, egalitarianism, and an increased participation of women and laity (McMahan 2012). Concepts such as rebirth and karma have been understood metaphorically and explained in psychological or symbolic terms so as to resonate with a more "modern" audience. Buddhist modernism has also popularized the secular practice of mindfulness meditation, which involves non-religious, therapeutic applications of Buddhist techniques. Mindfulness has been integrated into various domains such as healthcare, education, corporate settings, and psychology. It has been stripped of its explicitly religious elements, making it accessible to a broader range of people across different religious or

non-religious backgrounds. Moreover, Buddhist modernism has evolved in response to the needs and aspirations of individuals living in a modern, multicultural, and pluralistic society. McMahan (2008, pp. 1–2) recalls an experience where he witnessed Tibetan monks chanting and performing a short puja ceremony at a nightclub in Lancaster, Pennsylvania. Reflecting on this moment, he characterizes this encounter between Buddhists and interested Westerners as one which makes up the growing and shifting patterns of overlap between Buddhism and Western culture. However, it must be noted that Buddhist modernism is not a monolithic movement, and its manifestations differ across different cultural and regional contexts. It is a global network of movements that is not an exclusive product of one geographic or cultural setting (McMahan 2012, p. 160).

In the United Kingdom, Buddhist modernism has manifested itself in a variety of emphases, emerging as a significant development within the broader Buddhist landscape. First, Buddhist modernism in the UK has led to the integration of Buddhist principles and practices with psychology and therapy. Mindfulness-based interventions, such as Mindfulness-Based Stress Reduction (MBSR) and Mindfulness-Based Cognitive Therapy (MBCT), have gained popularity and are widely used in various secular settings, including healthcare, education, and workplace well-being programs (see Cook 2021). Secondly, the phenomenon has cultivated an eclectic and pluralistic approach to religion. Buddhist modernism in the UK embraces a diverse range of Buddhist traditions and teachings. There is a tendency to draw from various Buddhist schools, such as Theravada, Zen, and Tibetan Buddhism, and integrate them with contemporary ideas and practices from other spiritual traditions (e.g., Winkler et al. 2023). However, most pertinent to this research, Buddhist modernism in the UK has led to an increased emphasis on social action and engagement and the application of Buddhist principles to address social issues and promote justice. This was prompted by Western engagements with a variety of crises in Asia, namely the Vietnam War, and the inspiration they took from Asian engaged Buddhists (Henry 2013). Engaged Buddhists were incredibly active in struggles against the war. While some brought out family altars into the streets in an attempt to stop tanks from using the roads (see Wirmark 1974), others opted for the more radical tactic of self-immolation, or setting oneself on fire. These acts captured worldwide attention, specifically in the West. According to Meredith Kaufman (2013), Buddhism during the war became a fixation by Western media and, although their depictions were often wholly inaccurate, Westerners became captivated by the images of the religion that they were seeing (Kaufman 2013). As a result, the Antiwar Movement (also understood as the Peace Movement) took hold in the U.S. and the UK. In America, this movement grew into an unstoppable force, with leaders being pressured to leave Vietnam on moral and economic grounds (Dumbrell 1989). In the United Kingdom, British citizens followed suit, organizing actions such as the "March of Shame", where six thousand demonstrators congregated at Trafalgar Square to protest the Vietnam War (Thomas 2008). In both America and Britain, Buddhist responses to the war as well as Buddhist ideals of compassion and nonviolence inspired these movements and actions, which led to an increased interest in Buddhism's capacity for social engagement. This was ultimately one of the main factors leading to the success of engaged Buddhism in the United Kingdom, with large swathes of the British public calling upon what they saw as Buddhism's traditionally peaceful mindset, which was considered to be a powerful tool for change (Buyukokuyan 2011).[4] Therefore, it was Asian engaged Buddhist responses to the Vietnam War which signaled the beginning of the UK's interest in Buddhist forms of activism and led to the inculcation of engaged Buddhism into the country's religious tradition.

Today, engaged Buddhism has materialized in many forms. In Asia, engaged Buddhists provide services in the realms of elderly care (Aulino 2016), cleaning (Kuah-Pearce 2014), and disaster relief (Samuels 2016), among others, while Western engaged Buddhists, often in collective engaged Buddhist groups or broader activist movements, work to abolish the death penalty, safeguard nuclear waste, combat racism, misogyny, and militarism, and protect animals and the environment (King 2009). In the United Kingdom, the Karuna

Buddhist Vihara focuses on social engagement and compassion-based activism, supporting projects related to social justice, humanitarian aid, and community development. Similarly, the Network of Engaged Buddhists (NEB) is a network of Buddhist individuals and groups in the UK who are committed to applying Buddhist teachings to social issues, organizing events, workshops, and retreats that explore the intersection of Buddhism and activism. However, above all, arguably the main focus of engaged Buddhism in the UK and around the world is the state of the environment. Thich Naht Hanh's Plum Village Sangha sees engaged Buddhism as being at the forefront of environmental activism, and a great deal of academic attention has been paid to engaged Buddhist ethics in regards to climate change (see Batchelor and Brown 1992; Badiner 1990; Strain 2016; Javanaud 2020). One major reason for engaged Buddhism's vested interest in the environment is often attributed to the extensive relationship between Buddhism and nature. The Pali Tipitaka (also known as the Pali Canon or the Tipitaka) is a collection of scriptures in the Theravada Buddhist tradition which, although contentiously debated, is thought to have dated back as far as the 5th and 3rd centuries BCE and contains many references to the natural world. For example, it proclaims a need to care for the environment and living creatures. Similarly, in the Jatakamala, a retelling in Sanskrit of a selection of Jataka stories, the Buddha urges his followers to have sympathy for animals and that when it comes to avoiding suffering, all living things are the same. Therefore, one should not inflict suffering on any living thing, human or nonhuman, and the Buddha made a point to condemn activities such as animal sacrifices as being torturous and wasteful (Dhammika 2015). As a result, environmentally engaged Buddhist groups have formed all over the country, building on the relationship between Buddhism and nature to advocate for sustainability, counter consumerism and promote environmental healing.

XR Buddhists first emerged as a subgroup of Extinction Rebellion in 2019, a year after the movement's creation. The group originated from the now disbanded group known as the Dharma Action Network for Climate Engagement (DANCE) which sprung from the work of Buddhist teachers involved in the meditative retreat center in Devon, UK, known as "Gaia House". The aim of DANCE was to provide "a forum for the wider sangha to explore bringing Dharma responses to the climate crisis". One interlocutor, John, a founding member of both DANCE and XR Buddhists, told me of the group's history in one of our discussions. John was a middle-aged psychotherapist based in an area near Stansted Airport. With a long history as a Buddhist, he often brought his spiritual convictions to his counselling, specifically implementing mindful and meditative techniques. John also possessed a similarly long history with eco-activism, leading him to combine the two passions and dedicate his life to bringing Buddhist solutions and perspectives to the climate crisis. Today, John is incredibly active, especially in an organizational capacity, for XR Buddhists. He is well known for his actions in protest of Barclays Bank and their continued investments in fossil fuels, with one interlocutor even remarking to me that "they must have [John] on some Barclay's black list! Banks lock their doors when they see him coming".

At the time of DANCE's creation, John was involved in setting up a branch in London. He got together with another founding DANCE member, holding meetings in their homes in South London for an entire year, going back and forth on what they could actually do: "What could a good, proper Buddhist actually do?" he said, "there were endless discussions. . .thoughts on putting financial pressures which sounded dull, we held film showings, but ultimately, we felt completely rudderless." After a while of this back and forth, John came across an image which would change the entire course of the group. It was a photo of American activists sat in Washington D.C., dressed all in black, being dragged away by police. John became inspired by this image and brought it back to the group: "I said we could use this by bringing something we know how to do—meditation—to our activism." The group first began by meditating outside of a Shell building in South Bank, an activity that quickly became a regular occurrence, with numbers growing every time. DANCE became involved in the first ever XR protest by sitting in front of a Barclays

Bank. After this, the group joined in on the Rebellion in April of 2019, which led DANCE members to want to formally disband and "become part of the XR organism", as it were. The group became inspired by Extinction Rebellion's use of nonviolent direct action and decided that they needed to be more engaged or, as they put it in a posting on their website, more recognizably XR. In their invitation to join Extinction Rebellion Buddhists, the group stated that "We have found that Extinction Rebellion Buddhists has something unique to offer the Extinction Rebellion movement. Our compassion for all life, our capacity for equanimity and teachings of interdependence and impermanence have allowed us to offer a powerful presence, stability and a place of refuge to other members". Within the wider group of XR Buddhists, there are countless, smaller groups all over the UK, such as XR Buddhists Bristol, XR Buddhists Cambridge, XR Buddhists London, etc. Actions have been taken in parks, along roads, in the middle of streets, inside shopping malls and banks, even naked outside of King's College in Cambridge. Some of these resulted in XRB members, still in their meditative state, being dragged away and arrested by law enforcement. However, this form of protest has been conducted online as well as in person, with XR Buddhists conducting various meditations and faith vigils via Zoom. The power of meditation is a common sentiment among engaged Buddhists and is something which lies at the heart of this research. According to Adorjan and Kelly (2008, p. 48), for engaged Buddhists, meditation can be seen as a methodology of liberation which ultimately can affect social change.

With XR Buddhists emerging from the confluence between Buddhism and environmentalism, while the connection between Buddhism and the environment is continuously drawn upon, the relationship between Buddhism and social activism is not always so clear. Buddhism possesses an image in the West as a morally engaged tradition. In line with this thinking, many British Buddhists think their beliefs require them to engage in social activism. Such is the case with XR Buddhists, with many members telling me that their spiritual commitment to compassion and their belief in the interdependencies of all living beings required them to engage in societal suffering, to try and act out their Buddhist practice on a larger, more public scale. They are voicing a very common view of what many believe it means to be a Buddhist in the modern world, a world that is being increasingly affected by climate change, with engaged Buddhists across the UK and the globe combining their religious identity with and joining in environmental activism. However, many question whether the Buddha's teachings actually endorse social resistance and activism in any form, with some going so far as to argue that participation in these activities completely goes against these early teachings. As a result, Buddhism appears to be divided between "the value of responsible social engagement and the need for withdrawal from society to achieve individual liberation" (Nicol 2015, p. 61). In the United Kingdom, even though there has been a significant rise in Buddhist numbers since 1997, the numbers of Buddhists who identify or belong to an engaged Buddhist group are relatively small when compared with the wider Buddhist populace in the country. Phil Henry (2013) argues that over 90% of engaged Buddhists in the UK are Western converts. From these understandings, with an estimated Western Buddhist population of around 60,000, Henry gauges that only around 1.67% might identify themselves as an engaged Buddhist. With a fractional 24,000-person increase in the total population of Buddhists in the country from 2011, the time of his research, and 2021 (249,000 to 273,000), it is safe to say that this percentage still bears accuracy.[5] When discussing the history of XR Buddhists with me, John noted that many Buddhists associated with DANCE dropped out because "they thought Buddhists shouldn't be opposing anything, others thought we should be doing more". Therefore, in the section that follows, I aim to explore the juxtaposition between these two distinct factions, attempting to uncover whether the Buddha himself could be considered an "activist" of sorts, in terms of his desire for social harmony, and whether any support for activism can be found in his early teachings.

## 4. Was the Buddha an "Activist"? Understandings of Social Engagement in Early Buddhist Scripture

What we now refer to as "Buddhism" had its origins in the fifth century BCE in the Gangetic region of northern India, where Guatama Siddhartha, more commonly known as the Buddha, began teaching about his experience in attaining enlightenment (Gombrich 1988). From these humble beginnings, Buddhism has spread across the globe, albeit in a less than linear fashion and with the core teachings having undergone various regional- and cultural-level adaptations. However, let us begin at the beginning, with Guatama Siddhartha, who, according to later biographical works, had been born as the son of a king and queen. Although his father tried to shield him from the cruel realities of human existence, Siddhartha, as a young man, happened to witness various severe forms of human suffering: old age, illness, a dead body, as well as an ascetic renouncer. Siddhartha's exposure to these realities led him to the realization that all earthly and human pleasures are transitory, and that this impermanence underlies all human suffering. Leaving everything behind, Siddhartha became a renunciate, living in the forest and denying himself food until he was near starvation. At this point, however, he recognized that this path of denial to attempt to quell human desires actually increased, rather than alleviated, human suffering. Committing to meditate until he had discerned the solution to the problem of human suffering, Siddhartha stopped living as a renunciate, nourished himself, and sat beneath a bodhi tree in Bodhgaya, in the present-day state of Bihar in India. Through this period of intense meditation, Siddhartha, at age 35, is thought to have realized the Truth, the Dhamma, and attained enlightenment (Gombrich 1988). Through this process, Siddhartha became the Buddha.

Having accomplished his personal objective of solving the riddle of human suffering and attaining enlightenment for himself, the Buddha considered whether to share the information with others. Ultimately, though tempted to keep it to himself, the Buddha decided to share his realizations. The Buddha often stressed that it was "his own realization, his experience of the deathless state of nibbāna, which entitled him to teach, for he was teaching not a mere theory but a practice which he knew to work" (Gombrich 1988, p. 46). Consequently, before his death at 80, the Buddha founded an order, the Sangha, with the aim of preserving his teachings (Gombrich 1988).

Many see the Buddha's decision to spread his teachings in the first place as proof of his intention for social engagement and change. In the context of my research, the vast majority of my interlocutors held a steadfast belief that the Buddha himself was an activist, citing examples such as his allowance of women into the order, his condemnation of the caste system, his disapproval of war, etc. One such individual was XR Buddhist Erica. A native Bristolian, Erica was a spiritual counselor at an interfaith foundation, leading programs and "doing bits and pieces" in grief facilitation. Her mother left the Catholic Church when she was 10 years old and, along with her stepfather, began to practice Zen Buddhism. Through her mother, Erica also became interested in activism: "I've been an activist all my life" she said in a later interview, recalling her first "Save the Whales" Green Peace march at 11 years old. From there came engagements with the Campaign for Nuclear Disarmament (CND) as a teenager, with Erica becoming involved with the "Greenham Common Women's Peace Camp" at only 16 years old, where she joined thousands of other women in chaining herself to the fence of the base in protest of their use of nuclear weapons. Continuing the tradition of instilling her passion for social justice into her own children, Erica's family became increasingly engaged in the climate crisis throughout the years, becoming involved with Green Peace, Friends of the Earth, and Fridays for Future. Today, however, she tells me that her focus is more on "sacred activism" than civil disruption. When I asked her opinion on the Buddha's intentions for social change, she remarked, "The Buddha spent seven weeks on his own in the forest grove wondering whether he should go out and teach. He thought no one would understand. So he could have just kept it to himself. But he didn't. He went and taught, and he travelled around. He taught kings, a number of rulers he taught. So, how was he not interested in social transformation? Why was he interested in teaching

those people who were the leaders of the society? He recruited thousands of people into his Sangha. They were a force for social change by their very existence. He taught that there was no inherent distinction between different castes. I think the Buddhist teachings were socially very revolutionary." These arguments are backed up by many scholars of Buddhist studies. For example, Christopher Titmuss, a highly influential engaged Buddhist academic, explored how the discourses of the Buddha shed light on the daily lives of Indian citizens during his time and revealed how the Buddha outlined radical changes motivated by the desire to create a harmonious society. For example, by ordaining women into his Sangha, the Buddha enabled them to live a nomadic way of life, meaning that they now had another option to being wives, mothers, and servants.[6] Similarly, Battaglia (2015) argues that women played definitive roles in the religious tradition and that the original Pali literature was incredibly egalitarian. For example, in the canonical Samyutta Nikdy, the Buddha declared that "whoever has such a vehicle, whether it is a woman or a man, by means of that vehicle shall come to nibbana" (Feer 1973, p. 33). In fact, mention of the inability to practice Buddhism or to attain enlightenment due to physiological sex does not appear anywhere in Buddhist texts. The Buddha defined the term bhava as a process of "being" that is not metaphysical but produced by our minds. In the piece now known as the "Buddha Bhava," the Buddha described all sentient beings as being capable of enlightenment. Therefore, the Buddha proclaimed that both men and women were able to achieve salvation regardless of gender (Pipat 2007). In another Buddhist teaching known as the "Middle Path," the Buddha advocates for "transcending duality." Concepts such as man/woman were seen as "empty forms,"; therefore, the division of the genders was baseless (Pipat 2007). Overall, the Buddhist teachings mentioned above seem to emphasize striving for social equality between men and women, with the Buddha even going so far as to argue that gender is essentially irrelevant to Buddhism and the ultimate goal of salvation.

Moreover, the Buddha, as mentioned, was also very outspoken against the caste system. In one story, Christopher Titmuss recalls how Ambattha, a Brahmin, was sent to meet with the Buddha to test whether he was truly an incomparable religious teacher. Upon their meeting, Ambattha noticed that the Buddha permitted young people to sit on the highest seats, a privilege which was supposed to be exclusively reserved for elderly Brahmins. Ambattha lambasted the Buddha and his Sangha for refusing to show respect for the Brahmins and, in response, the Buddha traced the family history of Ambattha, showing his family to originate from a slave who had sex with Kanha, the illegitimate son of a King. Ambattha was overcome with shame and embarrassment. Out of compassion, the Buddha told him that Kanha had lived the life of a great sage with vast intellect and that he should be proud to be a part of the family of Kanha. The Buddha made it clear that it was the actions of a person that mattered, not where they came from.[7]

However, not all academics are so convinced of the Buddha's desire to bring about societal change. One thing that almost anyone will know about Buddhism is that compassion is one of its most important values. However, the line between a support for compassion and a support for social activism is not always so clear. While activists, particularly today, strive to utilize compassion to enhance their own lives, as well as the lives of others and the planet, Buddhist conceptions of compassion are primarily intended to be manifested in response to specific instances of suffering within local contexts. The emphasis within Buddhism is placed on displaying compassion in modest and unassuming ways, rather than through grand collective endeavors. As individuals traverse their own paths through life, they encounter instances of suffering among fellow beings and, in those moments, they extend their compassion through acts such as offering food, medicine, or simply providing companionship and reassurance. It is important to note that the Buddha does not characterize compassion as grandiose, sweeping actions that aim to benefit all beings and creatures (Faure 2003).

Faure (2003) goes back to the original Pali texts in order to combat the common arguments listed above which paint the Buddha as an activist. He reminds us that, something which is commonly ignored amongst engaged Buddhists and Buddhist academics, is the

fact that the Buddha was shown to be extremely reluctant to admit women into his order. When his stepmother requested that the Buddha allow women entry, it is said that he refused her three times before finally relenting. His reasoning for his reluctance was his fear that accepting women into the order "would bring about the decline of teaching" (Faure 2003, p. 23). Faure also brings up the Lotus Sutra, which claims that the only way for a female to achieve equal footing is to transform into a male (Faure 2003). Bartholomeusz (1994) adds to this discussion in her own book, referencing the Pali Mahavamsa, which details how Buddhists considered monks to be more worthy of support than nuns. Although the Mahavamsa acknowledged that nuns could attain enlightenment, Bartholomeusz (1994, p. 3) claims that that is the extent to which women are recognized, as most of the text focuses on the support of the order of monks by kings. Moreover, while the Buddha does claim that both men and women are able to achieve enlightenment, he hypocritically debases this statement with blatantly inegalitarian claims in other works. For example, the Buddha preached to monks about the dangers of women, and how they needed to be ever vigilant of women who may try to entrap them (Bartholomeusz 1994, p. 5). Therefore, whether implicitly stated or insinuated, Buddhist texts have often described the superiority of men and the inferiority of women, which blatantly contradicts the notion that gender is an "empty form" and calls into question his desire to instigate societal change via the equality of the sexes.

Moreover, Scherer (2019, p. 156) explores the often-ignored ableism which is prevalent in Buddhist scripture through "popular mono-causal reductionism of karma theory". She references Chapter 3 of the Lotus Sutra which, in English, translates to "Those who do not have faith in this my scripture, when they are born human again are then born idiots, lame, crooked, blind and dull" (3.122). The chapter continues with "[The blasphemer] foolish and deaf, does not hear the dharma" (3.129ab) and "when he obtains human birth he becomes blind, deaf and idiotic; he is a slave, always poor" (3.132a-c) (tr. Scherer 2019). In Chapter 28 of the Lotus Sutra (26 in Sanskrit), it is written that to defame the sutra results in visible ugliness. These quite blatant expressions of prejudice reveal contradictions in the Buddhist scripture as to the equality of all living beings, which is continuously referenced upon by engaged Buddhists as evidence for his desire for social change and a harmonious and just society. However, impairments in this religious discourse also "act as tools of persuasion and motivate by leading to the performance, in the Buddhist case, of good deeds" (Deegalle 2006, p. 16).

Overall, those who see the Buddha as a social reformer are at odds with those who recognize the multiple discrepancies in his teachings. Richard Gombrich (1988) argued that the Buddha himself had no desire for social change. His concern was to reduce human suffering and help people achieve enlightenment, not to reform the world. If anything, his preoccupation was in helping people leave the world. In his opinion, the Buddha never tried to reform societal inequality, he only declared its irrelevance to salvation (Gombrich 1988). Moreover, while the Buddha was aware of what one might call "social movements" by today's standards, he did not encourage his disciples to participate in them. This is because, social activism, he says, is in tension with Buddhist virtues. There are many qualities that Buddhists are meant to cultivate, like calm, tranquility, equanimity, gentleness. In many instances, Buddhist practice is likened to that of a gardener, as they too exercise these virtues. The gardener looks after their own little part of the world; they tend to it and take care of it. The gardener does not attempt to transform every part of nature in every part of the world, but they focus on their own small section of it (Gombrich 1988). This is why the Buddha advises, when questioned about politics, "Be islands unto yourselves, be your own refuge".[8]

These sentiments are echoed by Buddhists all over the world who disagree with the recent trends in "engaged Buddhism", and as a result of these teachings, throughout history, Buddhism has been continuously criticized for its other-worldly, enlightenment-focused tendencies and its blatant refusal to participate in social issues (Kuah-Pearce 2014). Therefore, in the United Kingdom, groups like XR Buddhists are in contention with

the majority of British Buddhists who seemingly disagree with their actions or refuse to participate. Many XRB members, who are also dually affiliated with more "traditional" Buddhist communities and centers, told me that the reason they chose to join XR Buddhists was because of the lack of social engagement that they were receiving within their own sangha. In Bristol, for example, Theo told me of their struggles with getting their sangha to engage in environmental issues: "I know there have been people in different sanghas who have really struggled to get their sanghas to engage", they began, "...when I moved to Bristol and started going to my sangha I was like 'why don't we hold a workshop around climate change?' and...lots of the people who I was speaking to were anti-XR, or if they were XR it was a thing they did outside of the sangha. And so...we really had a fight over what it was we were going to talk about and they basically didn't want us to bring anything remotely 'political' into the discussion...someone even sent me 14 mindfulness trainings and one of them was on kind of 'harmony in the sangha' and another was on 'not being political' but it also talked about engaging with compassion and reducing suffering in the world and it's kind of like, well, how do you do that if you're not engaging?". The issues that Theo discussed were vocalized by countless other XRB members. As a result, with Buddhist practice commonly associated with personal spiritual journeys, often considered private and inward, sometimes even selfish (Jones 2003), it is clear that the Buddha's early principles and their manifestations in the majority of UK Buddhists and sanghas are not compatible with modern-day activism.

## 5. Buddhist Constructions of Care and Dependent Origination as an Ethical Imperative

However, while Buddhism may not have a lengthy history in supporting social activism, it does have a deep and intimate relationship with care. "Care", for Fisher and Tronto (1990, p. 40), involves everything we do "to maintain, continue, and repair our 'world'...That world includes our bodies, our selves, and our environment" which operates in "a complex, life-sustaining web". Care conveys a sense of closeness, love, and concern (Besnier 2015). More than a service or a commoditized activity (e.g., healthcare or sex work), care is a social and emotional practice which depends on cultural expectations and norms (Alber and Drotbohm 2015). Stemming from the work of feminist scholarship, care is understood to have been continuously undermined due to its association with women. Care practices are often characterized as "unproductive" and professions which involve care work largely receive less pay and social prestige (Hakim et al. 2020). However, within realms of family, state, and market, we have witnessed a shift away from care "as a fundamental element of social organization" and toward care as a positive, productive activity "compensating for the hardships caused by new gendered and global inequalities" (Thelen 2015, p. 1). As a result, since the 2010s, anthropologists of care have worked to explore care not only as reactive but productive, transformative, and fundamental to social organization (Thelen 2015).

According to Buddhist literature such as the Vinaya Pitaka and the Sutta-Pitaka, one of the Buddha's focal concerns was on well-being. In fact, it was his experiences witnessing the impermanence of life through sickness, old age, and death that is said to have prompted the Buddha to reject pleasure and seek out a way to end the painful cycle of rebirth. The Buddha argued for the importance of mental health as the ultimate goal of human society and emphasized the need for people to address issues of attachment (moha), anger (raga), and delusion (dosa), which cause suffering (Hong and Kurane 2019). With suffering, from a Buddhist standpoint, considered to be caused by the mental habits of the sufferer (Gombrich 1988), Buddhist perspectives on care are often initially interpreted through the cultivation of the internal self, which is achieved, largely, through meditation (Hong and Kurane 2019).

Since its inception and up until today, meditation has been continuously used for mental and psychological support. But first, a definition of the practice is needed. As a somewhat elusive concept, it can manifest itself in both silence and sound. It encompasses

both stillness and movement (Pagis 2010). It has the potential to invigorate, enlighten, and heal (e.g., Myers et al. 2015), yet it can also evoke pain and exhaustion (Cook 2010; Pagis 2010). The practice is used across schools and therefore possesses a diverse array of techniques. For this reason, Keown and Prebish (2010, p. 514) define Buddhist meditation, broadly, as "a wide variety of techniques designed to produce heightened states of concentration and awareness that lead to knowledge, wisdom, and liberation". In the Theravada tradition, widely accepted as Buddhism's oldest existing school (Gombrich 1988), teachings on meditation have been central to Buddhist practice for centuries, with early texts steeped in discussions on the subject. For example, the Eightfold Path (ariya atthangika magga), an early summary of the Buddhist journey, is comprised of eight components: right view, right resolve (the wisdom section), right speech, right action, right livelihood (the morality section), right effort, right mindfulness, and right concentration (the meditative element) (Shaw 2021).

Meditation in early Buddhist practice was aimed at progressing one's spiritual development as well as promoting happiness and well-being. In this sense, meditation has been used all over the world, across time and space, as an opportunity for self-care. Self-care has been of increasing academic interest in recent years. Care underpins the meanings of personhood and poses questions of what it means to turn care onto the self. By centering care on the self, anthropologists explore the ways in which people care for themselves, what happens when people are unable to care for themselves, and how particular systems produce and value subjects (Rosenbaum and Talmor 2022). However, Buddhist interpretations and practices of care are not limited to internal well-being. The theme of care, especially for others, appears often in the Dhamma. In the Satipatthana Sutta, for example, the Buddha provides an assurance of awakening to those who properly practice the satipatthanas, i.e., those who care properly. The Ven. Analayo commented that the Pali word satipatthana can be translated as "attending (or caring) with mindfulness". Stephen Batchelor (2005) even posits that the Buddha's last injunction was for his followers to practice with care (appamada). As a result, Buddhist perspectives on care are often interpreted through a variety of principles, such as compassion (karuna) (Kuah-Pearce 2014) or karma and merit (Aulino 2016). Moreover, as Buddhist interpretations of care are not confined to the internal realm, neither is the practice of meditation. Michal Pagis (2015), in their article "Evoking Equanimity: Silent Interaction Rituals in Vipassana Meditation Retreats" characterizes the feelings of equanimity achieved in meditation as a practice of extending self-care outward. In meditation retreats, participants are together in a way which allows for "silent social attunement" (Pagis 2015, p. 39) which cultivates an equanimity with and for others. In these retreats, which can last for days or even weeks at a time, meditators share a space in almost complete silence. It is this silence that allows for a "nonverbal transformation of the self through the social production of a heightened emotional experience of equanimity" (Pagis 2015, p. 42). In a meditation hall, with individuals sitting in silence and close proximity, sharing the same activity, a "synchronized non-movement" takes place amidst moments of intense meditation. This results in actions such as coordinated breathing, "contagious relaxation", and the transfer of emotions from oneself to others like equanimity and peacefulness (Pagis 2015, p. 51).

Overall, sentiments on well-being and care have been instilled in Buddhism since the days of the Buddha himself. While often initially considered from the perspective of internal cultivation, with meditation continuously, throughout history, used as a practice for self-care and spiritual progression, Buddhist perspectives on care have always involved care for others as well. As a result, even with the juxtaposition between care for others and societal engagement being incredibly stark and conflicting, and with Buddhism, as a religion, seemingly at odds with what it actually means to be a "Buddhist" in the modern world, engaged Buddhist groups are collectively voicing the idea that perhaps in the 21st century it is time to further adapt Buddhist teachings for a globalized world, especially one that is increasingly being affected and transformed by universal issues such as climate change. As a result, XR Buddhists reflect the desire to extend meditation and its associations

with care outwards, to embody their religious practice and live it out in the world for others to see. In doing so, the group resists early interpretations of Buddhism and establish themselves as a relatively new force within UK Buddhism.

However, as Buch (2015) notes, in English-speaking regions, care connotes both affective concern (caring about) and practical action (caring for), meaning that care involves both feelings and bodily processes (Fisher and Tronto 1990). Building off of this, within discussions of the so-called "ethics of care", increasing attention is being paid to the ways in which care can be used for political critique. For XR Buddhists, it is the Buddhist principle of dependent origination that formulates the practices of Earthly care that they exhibit in meditation and acts as a politically charged ethical imperative that contests traditional Buddhist interpretations of social engagement.

> If...I believe that I am separate, then...I may feel rage, anxiety...hate...But if I understand [dependent origination] . . . I start to understand that the environment is me. The world is not outside of us, the world is us. And when we tap into that . . .we see we are the bankers, we are the people cutting down the forests, we are also the people living on Pacific Islands where the sea is rising. . .we are the ones profiting from the companies selling cash crops and fossil fuels and drilling for oil. We are also the animals, we are the oceans, we are all the interconnected, living systems of the planet. We cannot exist separate from those things, and we must show love and compassion and understanding to all of them.

These words were uttered by XRB Bristol member James. James identified himself as an "ex-anarchist". Now a pacifist and devout Tibetan Buddhist, as well as a massive proponent of nonviolent direct action, James reflected on his past life in a discussion we had during day two of the October 2022 Rebellion, where he used violence and anger to voice his opinions. Sitting on the grey flagstone steps in Trafalgar Square as we waited for the thousands of Rebellion-goers present to begin the procession towards 10 Downing Street, where the group as a whole would take over the area until 6 p.m. that evening, he explained to me how, through finding Buddhism and, more specifically, cultivating a deep understanding of dependent origination, his feelings of anger turned into feelings of care. These sentiments became a universal theme throughout my time with the group, leading me to conclude that, for XR Buddhists, the Buddhist principle of dependent origination (paticca samupada) (also understood as interdependent co-arising, dependent arising, interconnection, interdependence, or interbeing) is what dominates their understandings of care amidst the climate crisis.

Dependent origination, a key doctrine of Buddhist philosophy, the doctrine of causality, states that all phenomena arise and are dependent on one another (Boisvert 1995), existing in a complex web of inter-relationships which includes all living beings and the natural world. The basic principle is simplistic, with the Buddha describing it by stating "When there is this, that is. With the arising of this, that arises. When this is not, neither is that. With the Cessation of this, that ceases".[9] The belief of dependent origination is subject to a wide variety of interpretations, with early Buddhist texts characterizing it largely as a phenomenological or psychological principle referring to the workings of the mind and how suffering, craving, and self-view arise (see Shulman 2008). The principle is also closely connected with that of rebirth, and how rebirth occurs without a fixed self but as a process cultivated within specific contexts and relations (Williams and Tribe 2000, p. 64). Essentially, it provides a profound understanding of the nature of reality and the causes of suffering. According to the principle, everything in existence arises and ceases due to the coming together of various causes and conditions. It states that nothing exists independently or in isolation; instead, all things are contingent upon and influenced by multiple factors. This principle applies to both physical and mental phenomena and is often imagined via various links which form a chain of causation. These links are understood to form a cycle which perpetuates suffering and the cycle of rebirth unless it is broken through the cessation of ignorance and craving. Understanding dependent origination is crucial in Buddhism because it reveals the interconnectedness of all things and the causes of suffering. By

recognizing the interdependent nature of reality, one can develop insight into the transient and unsatisfactory nature of existence and ultimately strive for liberation.

With such a fixed focus on the attainment of enlightenment and freedom from suffering, these early Buddhist conceptualizations of dependent origination had no ties to social engagement. However, in reformulating the principle as an ethic of care within the context of the climate crisis, XR Buddhists apply the principle in order to deepen their understanding of the interconnectedness between human actions and the environment and foster a sense of responsibility to the suffering of the planet and all life. Such interpretations reflect the reformulation of Buddhist principles that has been encouraged by Buddhist modernism. As mentioned above, Buddhist modernism sought to present Buddhism as compatible with scientific and rational modes of thinking. In the UK, Buddhist modernism has fostered a greater emphasis on social engagement and activism and has subsequently reformulated principles, such as dependent origination, to encourage Buddhist practitioners to actively work for societal transformation, social equality, and environmental sustainability. Therefore, while early interpretations of dependent origination focused primarily on the causes and conditions that give rise to suffering and the cycle of rebirth, the reformulation of the concept by XR Buddhists goes beyond that, emphasizing the interconnectedness of all aspects of life, including joy, happiness, and the capacity for transformation and change.[10] It encourages the cultivation of compassion, responsibility, and a deep understanding to foster a sense of harmony and interconnectedness with all beings. While both concepts highlight the interdependence of phenomena, XR Buddhists place a greater emphasis on the transformative potential of this principle, the ethical obligations that arise from our mutual interdependencies, as well as its opportunity for worldly engagement as opposed to other-worldly enlightenment. The group employs the principle of dependent origination to help recognize the causes and effects of our collective actions on the natural world, encouraging a more responsible and compassionate approach to environmental stewardship. Returning once again to Fisher and Tronto's (1990, p. 4) definition of care as everything we do "to maintain, continue, and repair our 'world'" which includes "our bodies, our selves, and our environment" and operates in "a complex, life-sustaining web", XR Buddhists exemplify these understandings of care by using the principle of dependent origination to highlight the interconnected nature of all phenomena. In doing so, they emphasize that humans are not separate from the environment but rather deeply interconnected with it. Human actions and choices have a direct impact on the delicate balance of ecosystems and the well-being of all beings. Therefore, with the ecological crisis having an effect on every living entity, the state of the environment is our collective responsibility. Environmental degradation is thus considered to be enabled and supported by humanity's (specifically Western populations and their pervasive individualistic ideologies) ignorance to the interrelatedness of all things. In one interview with XR Buddhist Bristol member Andrew, the reformulation of dependent origination for these aims was made quite apparent. At the time of our interview, Andrew, a trustee for an international Buddhist organization, was in London for a round of meetings and, as a result, our discussion was conducted virtually. Dressed in a crisp white button-up shirt and grey rectangular-rimmed glasses, Andrew sat in an empty conference room as we discussed his motivations for participating in XRB meditative actions: "I think the main thing it...can teach, should teach...is basically all related to dependent origination...You cannot either turn your head the other way or put your head in the sand, or think, 'oh, that happening over there doesn't affect me anyway'. It all interrelates. So...do not think an action has no effect on you. You may think you are an individual but actually you are so influenced and interconnected with every living entity that it doesn't make any sense and doesn't reflect reality to not be involved". With care increasingly being used for political critique, and with academics and activists alike calling for a politics of care to be grounded in an understanding of our "mutual interdependencies" (Hakim et al. 2020, p. 23), XR Buddhists offer a particular ontological understanding of the world: that everything is connected and co-dependent (Gregory and Sabra 2008, p. 55). In doing so, the group offers a framework for ethical decision making, reformulating a princi-

ple which previously had no ties to social engagement as an ethical imperative, confronting Buddhism's other-worldly, enlightenment-focused tendencies and instead emphasizing the need for our interconnected reality to find expression in present-day action. Action, for XR Buddhists, takes the form of collective, public meditation.

## 6. From Conceptual to Embodied Knowledge: Meditation-as-Care

On 15 October 2022, I made my way from London Paddington to the Tate Modern Museum, like I had so many times before, for the October Extinction Rebellion protest. The atmosphere is almost festival-like. People are dressed in costumes, from fantastical tree spirits to eccentric tea party goers, extravagant displays and decorations were adorned on statues and walls, and the movement's trademark samba band is already in full swing, playing loud and rhythmically throughout the day's proceedings. The plan for the day is to take over the area surrounding the Tate. There will be speeches read out over a loudspeaker, outreach actions performed with the intent of engaging with the public, and demonstrations held, such as the burning of electric bills in protest of rising energy costs. However, XR Buddhists will not be engaging in these activities. Through the sea of neon flags and placards, I soon spot sixteen XRB members sitting along the riverfront near Millennium Bridge. After a few moments, a chime rings out. Each person finds a comfortable position before closing their eyes and beginning to meditate. Some sit on their knees, the ground beneath them, or on meditation cushions or chairs. Their positions vary slightly as well, with some practitioners clasping their hands together, others placing them on their legs or the ground beneath them. The group's chests rise and fall slowly and methodically. Their eyes do not open. For an hour, they will sit quietly with as little movement as possible, trying to focus on the bodily sensations they are experiencing before another chime signals an end to the day's action (Figure 3).

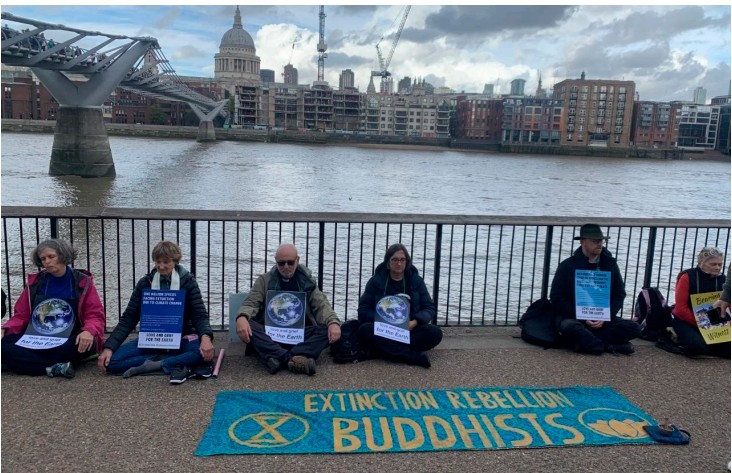

**Figure 3.** Photo by the author, 15 October 2022.

According to Michal Pagis (2010), while a conceptual understanding of Buddhist principles is crucial, it is only through meditation that these concepts are experienced on the bodily level and thereby "realized" as truth. In her research on Vipassana meditation retreats in Israel and the U.S., Pagis (2010) explored practitioners' experiences with painful meditation caused by long hours of sitting and maintaining the same position. She concluded that, from these experiences, the Buddhist principle of suffering (dukkha) becomes an embodied reality as participants realize how their body is attached to comfort and, consequently, how attachment is the root of all suffering (Pagis 2010). As a result, from the bodily sensations experienced in XRB's meditative actions, a conceptual understanding of the principle of dependent origination, reformulated as an ethic of care in the context of the climate crisis, becomes an embodied reality. Similar arguments are reflected in Buch's (2015) illustration of care as being composed of both caring feelings and caring actions, mentioned

above. Building off these understandings, XR Buddhists implement meditation as the vehicle by which Buddhist principles become inculcated in both the bodies and minds of practitioners and formulate an embodied ethic of care which the group implements in their direct engagements in the climate crisis.

Meditation dominates every aspect of XR Buddhists' practices, with the physicality of meditation and the embodiment of dependent origination being of central importance to all activities. Therefore, as XRB Cambridge member Murphy described in his interview, meditation serves as a "body awareness exercise". Murphy was an Irish-born journalist and Buddhist writer who played a prominent role in instigating meditative actions in Cambridge. Telling me of his personal experiences in meditation actions and how he was able to cultivate a deep understanding of the oneness of self and environment, Murphy said, "It's feeling the contact between my feet and my legs and the ground" he said, "feeling that connection. . . and feeling the emotional tone in my body. So, starting with physical sensations. . .and then, what's the emotional tone? Feeling that care in my heart. So basically, legs, feet, body, heart." This awareness "allowed me to recognize that my body is in constant interaction with the world around me", Murphy said, "from the ground beneath my feet to the air above my head, I am always touching the Earth".

Understandings of touch are informed by phenomenological perspectives on our corporeal "being-in-the-world" and the notion of intercorporeality (Merleau-Ponty 2010). Intercorporeality emphasizes the embodied and relational nature of human existence, referring to the idea that our lived experience and understanding of the world is fundamentally shaped by our bodily interactions and the shared bodily presence with others. According to Merleau-Ponty (2010), bodies are not merely objects in the world, but rather dynamic and engaged agents that actively perceive and make sense of the world through their interactions. Touch is a tool by which bodies achieve this, in which they interact with the environment and each other (Finnegan 2014, p. 194). The most pertinent aspect of this research, however, is elucidated by Maria Puig de la Bellacasa (2017), who characterizes touch as an intimate care practice, one capable of turning caring feelings into action. Although the touch achieved in meditation is quite different from the touch Bellacasa explores with farmers' handling of soil, it remains a crucial component of XRB's meditative practice. Meditators touch the ground they sit upon, they touch themselves, they touch the Earth with their breath, and the Earth touches them via wind, rain, sunlight. These sensations affect the body, creating a reciprocal connection and helping to formulate the practice of Earthly care that XRBs exhibit. Therefore, in line with Murphy's description of meditation as a "body awareness exercise", I elucidate XR Buddhists' intimate engagements with reciprocal touch via two commonly used tactics that I have witnessed across the group's actions and practices: breath awareness and body awareness.

"I am a part of the Earth", on the inhale, "the Earth is a part of me", on the exhale: this is a breathing mantra that XR Buddhists regularly implement in their meditative actions. During meditation, participants breathe deeply and slowly, a tactic which aids in focusing on one's breath entering and exiting the body. From breath awareness, XR Buddhists contend that they are able to develop a physical understanding of how their body intertwined with the world around them, helping to cultivate an understanding of one's interdependencies with the environment by fostering a direct and experiential connection with the natural world. Interlocutors often referenced how, by becoming aware of the air entering and leaving their lungs, they were able to sense their interconnectedness between their breath and the atmosphere, with one XR Buddhist even mentioning how, in focusing on this bodily function, she was able to foster an experiential understanding of how her breath intertwined with the air and subsequently the oxygen which produced plants and sustained all life on Earth. This recognition deepened her appreciation for the interconnected web of life and how one's well-being is profoundly dependent on and interconnected with the well-being of the environment. Merleau-Ponty (2010) argues that breathing is not confined to the boundaries of our skin but extends beyond, merging with the surrounding atmosphere and environment. In this sense, breath serves as a

medium of contact and communication between the body and the world. Merleau-Ponty (2010) highlights that, through breath, we constantly exchange and share the air with our surroundings, blurring the distinction between our individual bodily boundaries and the external environment. Therefore, it is this intermingling of breath, where one's breath touches the air and is touched by the air simultaneously, that is a core focus of XRBs' meditative practices which cultivate a sense of interconnectedness and intercorporeality amongst members, where one's bodily existence becomes inextricably entangled with the world.

As for the second tactic, in my observation of meditative actions I was continuously drawn to the ways in which XRB members would position themselves, consistently placing their hands on their legs or the ground beneath them. As their practice continued, their placements never moved and remained firmly rooted at all times. The aim of doing so, I was later told, was to focus on certain points of the body and how they were connected. This is a technique which every XR Buddhist I spoke to implemented in their practice, as it fostered a direct and experiential understanding of one's physical and energetic interconnections. In directing one's attention to specific points of the body during meditation, such as the feet, hands, or legs, participants were able to cultivate a heightened sense of embodiment. Interlocutors continuously spoke of becoming more aware of the physical sensations that they were experiencing at each point of their body, and how this awareness produced a heightened cognizance of the interplay between one's body and the environment.

At a January 2023 action organized by John, I was provided with clear evidence of the significance of reciprocal touch between body and environment. This action specifically targeted Barclays Bank and their continued investment in fossil fuels. The intention was to split into two groups, with the smaller one, comprised of three XR Buddhists, meditating inside the bank, and the other nine individuals meditating along the building outside of it. XR Buddhists' trademark flag would be spread out on the ground in front of them and a banner would be hastily stretched along an iron gate which read "Barclays Funding Climate Breakdown" (Figure 4). Listening to John's instructions for meditation practice, he said: "Start by bringing your awareness to your body. Feel the connection between the soles of your feet and the ground. Feel the softness of the grass beneath your feet, its coolness and vitality seeping into your body. Become aware of the solid connection between your feet and the Earth, sensing its profound stability and support. Feel how the Earth supports your weight…feel that weight pressing down into the Earth and feel the pressure coming the other way, that feeling of being held in your place". By no means an experienced meditator, my fieldwork experience thus far had been plagued by instances of a wandering mind, of desperately trying to focus my attention. However, with a knowledgeable and compassionate guide to offer support, I was provided with clear instructions and guidance on how to deepen my understanding of dependent origination: "Allow your awareness to expand…get an awareness of your whole body…just feel that sensation of being held by the Earth and the world around us. Of being here, right now." From John's advice, it became abundantly clear that the associations of touch and a focus on specific points of the body during meditation were what made one's interdependencies with the natural world an embodied experience.

Through meditation and its associations with touch, XR Buddhists retrain their "relationship to the senses" and are able to "perceive religious tenets in his or her self-identity" (Cook 2010, p. 108). In transforming their embodied subjectivity, the group recognizes the Earth's living presence and acknowledges their reciprocal relations with it, which intensifies "a sense of the co-transformative". Through the group's intimate engagements with touch, they cultivate a care practice based on the establishment of "close relationality" where "boundaries between self and other are blurred" (Puig de la Bellacasa 2017, p. 96). During meditation, practitioners touch the ground they sit upon, they touch themselves, they touch the Earth with their breath, and the Earth touches them via air, rain, sunlight, soil, etc. These are all experiences which humans experience on a daily basis. However, in placing dependent origination at the forefront of their meditative action, a practice associated with

the production of heightened states of concentration and awareness, the physical sensations experienced help to formulate the practice of Earthly care that XRBs exhibit. Thinking touch-with-care emphasizes an "intra-active reversibility" and cultivates a deep sense of "immersion" (Paterson 2006, p. 699). It is in this sense that XR Buddhists embody their "mutual interdependencies" (Hakim et al. 2020, p. 23), making dependent origination a lived reality and fostering an affective, intimate care for the Earth.

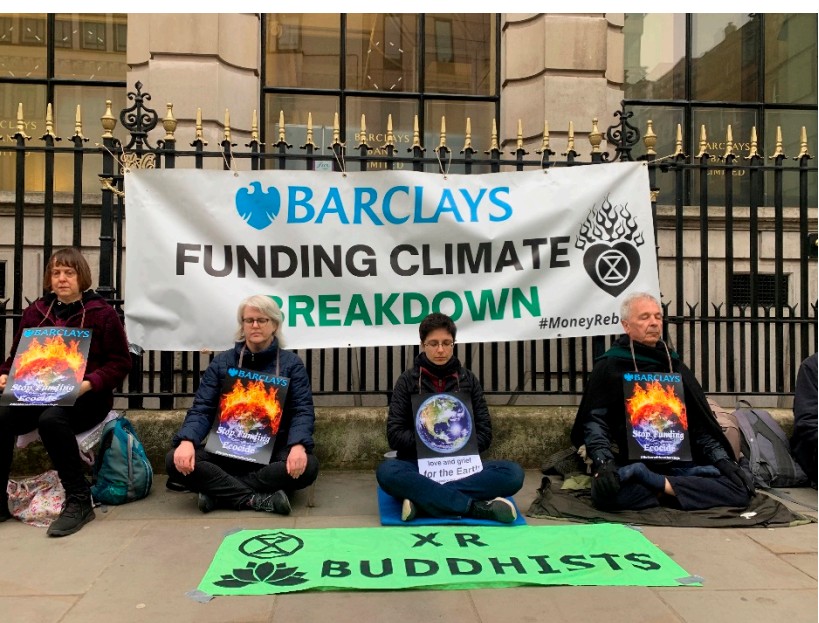

**Figure 4.** Photo by the author, 15 January 2023.

It has been proposed that current practices of meditation have undergone a shift in their ethical significance (McMahan 2008), becoming therapeutic instruments which cater to a population seeking solace from their discontent with the prevailing social and political order (Zizek 2005). Similar to the continuous undermining of care mentioned above, academics have also suggested that spiritual practices, at large, do little in terms of resisting the status quo and instigating social change, with meditation specifically critiqued for producing political passivity and a "regression" to a Hindu mindset (Gleig 2020, p. 773). This paper aims to contest these claims, painting XRB's meditation-as-care as productive, transformative, and a significant form of resistance, acting as a spiritually driven intervention for contesting the societal structures which have worked to produce the current climate crisis. According to Hobart and Kneese (2020, p. 2), care, when mobilized, "offers a visceral, material, and emotional heft to acts of preservation that span a breadth of localities: selves, communities, and social worlds". As a result, care presents an otherwise, an alternate mode of being which offers a way forward in the face of violence, economic crisis, and, most pertinent to this research, impending ecological collapse. Upon conclusion of the meditation, John addressed the group and said, "Think of this not only as an action, but a practice action. This is an opportunity to embody your Buddhist practice and live it out in the world for others to see." In taking meditation out of the sangha and onto the streets of the UK, XR Buddhists draw attention to the declining state of the environment, promoting their religion's emphasis on interdependence as a means of addressing ecological challenges. It is in this sense that the group's care practices illuminate the vulnerable attachments that provide means of survival for living beings (Allen 2021), aiming to dissolve the "I-it" relationship between people and planet and reanimate our affective engagement with the world (Holbrook-Smith 2017). Moreover, in encouraging practitioners to actively acknowledge and address the suffering and injustices faced by the planet, presenting an opportunity to intimately care for the Earth and express one's concern for it, the group challenges the common Buddhist tendency to retreat into personal

spiritual pursuits alone. XR Buddhists therefore reformulate meditation as a mere tool for individual liberation into a transformative practice capable of inspiring social change.

Puig de la Bellacasa (2017, p. 173) explores the "tensions in the temporal atmosphere", where images of a gloomy environmental future compress the present time for action through its urgency. This constant state of crisis means that the present is "diminished, mortgaged to an always unsure tomorrow". In the United Kingdom, this reflects an all too real reality, with Extinction Rebellion sending an urgent message that we are in a climate emergency and need to act now (before we go extinct), implementing disruptive tactics (e.g., blocking roads) with the aim of initiating swift societal change. Even the group's symbol, an encircled hourglass, conveys their "apocalyptic views" (Joyce 2020, p. 1). However, intimate practices of care can disrupt and suspend the future (Puig de la Bellacasa 2017). Meditation is about awareness of the present, about "remembering" where you are (Cassaniti 2018), with techniques implemented to achieve deepened awareness and understanding. These practices associated with meditation are therefore critical in this time of panicked urgency, as they challenge XR's "restless futurity" (Puig de la Bellacasa 2017, p. 175), making time for the experienced present which thereby extends the temporal landscape and time for action. As such, XRBs' use of meditation makes time for affective engagement with the world, contesting XR's future-driven imaginaries and subsequently utilizing care as a means for transforming meditation from a largely individualistic means for spiritual growth into a productive, effective practice of societal involvement which acknowledges the interconnected reality of environmental suffering that has been facilitated by modern structures and institutions.

## 7. Conclusions

According to Michal Pagis (2010), Buddhist philosophy emphasizes one way of knowing: wisdom (panna, in Pali). In her view, "this form of knowledge or wisdom is frequently regarded as non-dualistic, being in the realm of the body and the mind simultaneously". As a result, Pagis perceives Buddhist conceptualizations of wisdom as being constituted by two kinds of epistemologies: the conceptual knowledge of the tenets the Buddha taught and the embodied knowledge of said tenets gained in meditation practice (Pagis 2010, pp. 474–75). Therefore, it is through associations between mind, body, and the external world that XR Buddhists formulate an intimate practice of care for the Earth that is based on the interconnected reality of all life.

The subjective dimension of care plays a crucial role in shaping perception and interpretation, acting as a tool to resist normative forces imposed on an individual and carving out spaces for autonomy and self-expression. Engster and Hamington (2015) argue that care ethics can challenge traditional moral frameworks and address systemic injustices. In the context of this research, reformulating dependent origination as an ethic of care allows XR Buddhists to bridge the gap between the individualistic spiritual practice emphasized in traditional Buddhist literature and collective social action that has been encouraged as a result of Buddhist modernism's proliferation in the United Kingdom, arguing that personal transformation and ethical conduct must extend beyond the boundaries of the self and encompass the well-being of the entire ecosystem. By integrating environmental care as an ethical imperative into their subjectivity, XR Buddhists challenge the traditionally held notion, in the UK and elsewhere, that Buddhist practice should focus solely on individual liberation and detachment from worldly concerns. It recognizes that the path to liberation includes caring for and respecting the Earth, as it is an essential component of the interconnected web of existence. In this sense, caring for the environment is not separate from spiritual practice, but an integral part of it. As a result, in cultivating an ethic of care that challenges dominant norms set forth by early interpretations of Buddhist scripture, XR Buddhists actively shape their own subjectivities, resisting mechanisms of power and asserting their own values and ways of being that align with the current developments of Buddhist modernism that are occurring in the UK.

Moreover, within the inward-looking, enlightenment-focused tendencies of traditional Buddhism, meditation, through its enhancement of concentration and awareness (Keown and Prebish 2010), is often described as nothing more than a tool to further one's path towards awakening. However, as McMahan (2008, p. 184) argues, Buddhist modernism has radically transformed the traditional meaning of meditation: "Rather than exclusively a means of achieving awakening in a traditional sense, it [meditation] has. . .been reconfigured. . .outside of doctrinal and sectarian formulations". Today, religious practice takes place within "those moments in social life when the customary, given, habitual, and normal is disrupted, flouted, suspended and negated, when crises transform the world from an apparently fixed and finished set of rules into a repertoire of possibilities. . ." (Jackson 1989, p. 20). No such crises are more obvious today than the ones posed to our collective environment. Leaning in to Buddhism's deep relationship with care and well-being, XRB's meditation-as-care therefore represents a re-evaluation of the traditional Buddhist ideal of withdrawn, inward practice, emphasizing the importance of active engagement in reciprocal relations with our external world and the need for our "mutual interdependencies" (Hakim et al. 2020, p. 23) to find expression in action.

Gombrich (1988) points out that flexibility is in the spirit of Gotama Buddha. The Buddha himself compared his own doctrine to a raft. Just as one makes a raft to cross a river, only a fool, once having crossed, would carry the raft further. Gombrich takes this sentiment as meaning that the Buddha's preaching was to meant take people across the ocean of their spiritual journey; once they were across, they could go their ways without clinging to his words (Gombrich 1988). It is in this spirit that Buddhist modernism has encouraged teachings and practices to be adapted to modern contexts. In the United Kingdom, this pertains to an increased emphasis on the need for societal engagement. According to Sandra Bell (2000, p. 418), "Buddhism in Britain has moved beyond the initial period of transmission and institutionalization. Engagement with social and political realities reflects a new confidence and maturity." It is in this sense that XR Buddhists have left their rafts behind, following the trends of Buddhist modernism in the country and offering individuals a means to navigate the complexities of life in the Anthropocene by promoting a sense of collective responsibility for the state of our interconnected world.

**Funding:** This research received no external funding.

**Institutional Review Board Statement:** The study was conducted in accordance with the Declaration of Helsinki and approved by the School of Anthropology and Museum Ethnography Research Ethics Committee (SAME REC) in accordance with the procedures laid down by the University of Oxford for ethical approval of all research involving human participants (SAME_C1A_20_074; 24 July 2020; SAME_C1A_21_101; 30 September 2021).

**Informed Consent Statement:** Informed consent was obtained from all subjects involved in the study.

**Data Availability Statement:** Per the IRB protocols for this research project and my agreements with the organizations in this study, all field notes, interview transcripts/notes, and other data are private and stored on a password-protected device in the author's possession.

**Conflicts of Interest:** The author declares no conflict of interest.

## Notes

[1] https://www.parallax.org/mindfulnessbell/article/dharma-talk-history-of-engaged-buddhism/ (accessed on 22 September 2022).

[2] Some individuals and groups clearly belong at the core of the movement, such as Thich Nhat Hanh and his Plum Village sangha, and others are borderline, such as groups like Soka Gakkai, which place loving-kindness (kosen rufu) at the center of their practice but largely avoid what they view as "political" engagement.

[3] This is not to say that the West did not engage with Buddhism until the 19th century. In fact, evidence suggests that Buddhism spread westwards, at least into Iranian territory and the Graeco-Roman world, centuries before (see Seldeslachts 2007).

[4] However, it must be noted that, although the Antiwar Movement in America and the UK was influenced by and aspired to nonviolence, this was not always the case, and many instances resorted to violent tactics (see Dumbrell 1989).

5       www.ons.gov.uk/peoplepopulationandcommunity/culturalidentity/religion/bulletins/religionenglandandwales/census2021 (accessed on 24 July 2023).

6       https://www.christophertitmussblog.org/the-buddhas-campaign-to-change-society-part-one (accessed on 3 July 2023).

7       https://www.christophertitmussblog.org/the-buddhas-campaign-to-change-society-part-one (accessed on 3 July 2023).

8       https://www.accesstoinsight.org/tipitaka/sn/sn22/sn22.043.wlsh.html (accessed on 2 July 2023).

9       https://www.buddhistinquiry.org/article/dependent-origination/ (accessed on 2 July 2023).

10      This is not to say that XR Buddhists are the only engaged Buddhist group to reformulate the principle of dependent origination for environmental aims. The principle is implemented amongst a plethora of engaged Buddhist communities and practitioners. For example, Thich Nhat Hanh's (1987) concept of interbeing has also been reworked from the core sentiments of dependent origination. From the understanding that we are not separate entities, but rather part of a vast network of relationships with other people, animals, plants, the Earth, and the cosmos, interbeing calls for a shift in consciousness from a sense of separation and individuality to one of connection and unity. The principle invites us to recognize the suffering and well-being of others as our own, and to act with compassion and mindfulness in our relationships and in our actions towards the world (Nhat Hanh 1987).

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
