# Peer review of "Contesting Religious Boundaries with Care: Engaged Buddhism and Eco-Activism in the UK"

_religions, doi:10.3390/rel14080986_

Round 1

Reviewer 1 Report

Overall the manuscript is in good shape. It presents a case of Buddhist movement in UK that using a regular practice of Buddhist meditation in the public spaces on those crowded towns to urge public awareness of modern environmental crisis. Two comments:

1) The author could present much more general background for the idea of engaged Buddhism. Engaged Buddhism has emerged in some specific locations and contexts. In other word, this particular modern Buddhist movement is more popular in the Buddhist communities including Sri Lanka, Thailand, Vietnam, and Han Chinese communities. In some other communities, for instance, Tibetan Buddhism, it has a long tradition of very much engaged in this worldly affairs (including political and economic affairs) comparing with other Buddhist traditions.  

2) The autor’s presentation of the idea that XR Buddhist meditation extend the personal body to outward universe seems to be quiet repetitive. Reading through the manuscript, one can encounters many times of this.  I would suggest to discuss this idea intensively in one place of manuscript and leave some space for other things. One idea that author could explore is how the Buddhists’ engagement in the social affairs in this particular case is different from other form of engagement in other communities.  Author may could formulate something out of these differentiation by exploring the way in which the mass civil non-violence meditation disrupt the public awareness of their relationship with world they live.  I see the fact that XR Buddhists connects their religious teaching to the ecological concerns, but I did not see what their engagement mean to the structural changes and theoretical implication, for instance, the power relationship, capitalism, institutional arrangement, gender relationship and so forth. One question could be asked is: does the idea of care mean some type of structural changes?  For instance, is the concerns expressed through the meditation in the public space mean any changes in the agency that humans and non-human powers (let’s say transcendental power like karma) play in the human-environment (human-ecological) relationship? Or, another question: in thinking of the modern encounter of the Buddhist transcendental narratives and science-based social and institutional intervention, is the meditational protest a spiritually driven intervention to the current social order? Or, is it a secularly driven “social” movement?  Or both? Anything like that could be explored more.

Author Response

Reviewer, 

Thank you very much for taking the time to review my paper and providing such helpful notes. 

In regards to point 1: I have added slightly more detail on the history of engaged Buddhism in other Asian countries, using Tibet as an example. 

In regards to point 2: I have attempted to make this point less repetitive by more clearly marking the point where I discuss the significance of XRB's meditation-as-care in terms of external structural change.  By doing this, I aim to split the analysis of the group's meditative practices into 3 distinct realms: the subjective, the bodily, and the external. I also make clear that XR Buddhists' meditative protest is spiritually driven as opposed to a secular social movement.

Reviewer 2 Report

The author of the article is based on two years of ethnographic research the ways in which an environmentally engaged Buddhist group known as “Extinction Rebellion Buddhists” adapt their religious beliefs and practices in response to the challenges posed by the Anthropocene. This development in the Buddhist tradition referes  as “Engaged Buddhism” (“Applied Buddhism” or “Buddhist activism”). I believe that this study contributes to the research on environmentally engaged Buddhism in the United Kingdom. Ethnographic data was gained from interviews and participant experience. The author himself participated in the described realms of activism of XR, but despite this tries to use a critical scientific approach. Perhaps, as a wish to the author, some statistical sociological data could be given so that the reader has an idea of the social, demographic and other characteristics of the phenomenon under consideration. The author points out that in the United Kingdom, groups like XR Buddhists are in contention with the majority of British Buddhists because XR Buddhists extend meditation outwards, to live their religious practice out in the world. The author explains this by saying that for XR Buddhists, it is the Buddhist principle of dependent origination that formulates the practices of Earthly care that they exhibit in meditation and acts as a politically charged ethical imperative. In reformulating the principle as an ethic of care within the context of the climate crisis, XR Buddhists apply the principle in order to deepen their understanding of the interconnectedness between human actions and the environment. The author notes that this is not to say that XR Buddhists are the only engaged Buddhist group to reformulate the principle of dependent origination for environmental aims. However, it is not clear from the article which other groups practice this approach.

Author Response

Reviewer,

Thank you for taking the time to review my work and providing such helpful comments. 

In regards to the first point: I have added some statistical data so that the reader may have a better idea of Buddhist numbers in the U.K. as well as how small the number of engaged Buddhists in the country are in comparison to this population. 

In regards to the second point: I have added a few sentences in the notes section at the end of the paper which details Thich Nhat Hanh's concept of "Interbeing" as an example of a group/individual other than XRB reformulating the principle of dependent origination for environmental aims.